# The Efficacy of CPAP in Neonates with Meconium Aspiration Syndrome: A Systematic Review and Meta-Analysis

**DOI:** 10.3390/children9050589

**Published:** 2022-04-21

**Authors:** Carlos J. Toro-Huamanchumo, Maryori M. Hilario-Gomez, Nelson Diaz-Reyes, José A. Caballero-Alvarado, Joshuan J. Barboza

**Affiliations:** 1Facultad de Ciencias de la Salud, Escuela de Medicina, Universidad César Vallejo, Trujillo 13007, Peru; toro2993@hotmail.com; 2Sociedad Científica de San Fernando, Facultad de Medicina Humana, Universidad Nacional Mayor de San Marcos, Lima 15081, Peru; maryorimhg010514@gmail.com; 3Escuela de Medicina, Universidad Peruana Unión, Lima 15081, Peru; fideldiaz@upeu.edu.pe; 4Escuela de Postgrado, Universidad Privada Antenor Orrego, Trujillo 13007, Peru; jcaballeroalvarado@icloud.com; 5Unidad de Revisiones Sistemáticas y Meta-Análisis (URSIGET), Vicerrectorado de Investigacion, Universidad San Ignacio de Loyola, Lima 15081, Peru

**Keywords:** CPAP, meconium aspiration syndrome, newborns, systematic review

## Abstract

Background: In neonates with meconium aspiration syndrome (MAS), continuous positive airway pressure (CPAP) may be more beneficial compared to endotracheal intubation (ETI). We evaluated the efficacy of CPAP in neonates with MAS. Methods: Four engines were used to search randomized clinical trials (RCTs). We used relative risk (RR) and mean difference (MD) with 95% confidence intervals (95%CI) to assess the effect on dichotomous and continuous outcomes, respectively. In addition, we used the Paule–Mandel (PM) random effects model due to the anticipated lack of events. Results: Three RCTs were included (*n* = 432). No significant difference was found in mortality (RR = 0.82; 95%CI = 0.54–1.25; I^2^ = 71%; *p* = 0.36), need for ventilation (RR = 0.49; 95%CI = 0.15–1.56; I^2^ = 71%; *p* = 0.57), and incidence of pneumothorax (RR = 1.24; 95%CI = 0.30–5.12; I^2^ = 0%; *p* = 0.77) in the CPAP group compared to the ETI group. Regarding secondary outcomes, compared to the ETI group, no significant differences were found in APGAR at one minute (MD = −1.01; 95%CI −2.97 to 0.94; I^2^ = 98%; *p* = 0.31), APGAR at 5 min (MD = −1.00; 95%CI = −2.96 to 0.95; I^2^ = 99%; *p* = 0.32), days of hospitalization (MD = −0.52; 95%CI = −1.46 to 0.42; I^2^ = 94%; *p* = 0.28), and cord pH (MD = 0.003; 95%CI = −0.01 to 0.02; I^2^ = 0%; *p* = 0.79). Conclusions: In patients with MAS, there is no significant effect of CPAP use compared to ETI on primary, specifically on mortality, need for ventilation, the incidence of pneumothorax, and secondary outcomes.

## 1. Introduction

Meconium amniotic fluid aspiration syndrome (MAS) is one of the most common causes of respiratory distress in term and post-term neonates [1]. It is associated with atelectasis, hypoxemia, hypercapnia, persistent pulmonary hypertension, inflammatory changes, and inactivation of pulmonary surfactant [2]. Neonates with MAS have up to ten times higher risk of death than patients without MAS, and mortality from this disease has been reported to be as high as 26%. Some indicated therapeutic procedures are oral cavity aspiration and primary resuscitation measures if necessary [3].

Pharyngeal suctioning and tracheal suctioning improve the prognosis in neonates but do not eliminate MAS [4]. These procedures are usually painful and can cause pulmonary and systemic hypertension, bradycardia, intracranial hypertension, and hypoxia in the neonate, which are events associated with mortality [5]. On the other hand, it has been suggested that continuous positive airway pressure ventilation (CPAP) may be more effective than ETI in neonates with MAS [6]. The initiation of CPAP could further decrease the rate of ETI in the delivery room and the duration of mechanical ventilation with the successive benefit of reducing mortality and bronchopulmonary dysplasia without a significant increase in severe interventricular hemorrhage [7]. 

A few randomized controlled trials (RCTs) have been conducted to evaluate the efficacy of CPAP versus ETI in neonates with MAS. However, the evidence has not been synthesized. Such a synthesis could aid evidence-based clinical decision-making. Because of this, our objective was to conduct a systematic review to compare the efficacy of positive pressure of ventilation (PPV) by CPAP or ETI in neonates with MAS.

## 2. Materials and Methods

### 2.1. Registration

The protocol of the systematic review was previously published in PROSPERO (CRD42018104709). Significant modifications were made in this manuscript with respect to the previous protocol. These changes were mainly directed at the main and secondary outcomes approached. The update was due to the subsequent evaluation of the evidence, which allowed the outcomes evaluated in each study to be appropriately stated. These changes did not affect the development of the protocol or the evaluation of the efficacy of the treatment compared to the control. The recommendations of the PRISMA group (Preferred Reporting Items for Systematic Reviews and Meta-Analyses) were used to report this systematic review [8]. 

### 2.2. Search Methods

We searched for potential studies in Pubmed, Scopus, Web of Science, and EMBASE until 17 January 2022. We assessed randomized controlled trials (RCTs), including infants with MAS, with CPAP and ETI as the intervention and comparator, respectively. There were no restrictions on language or publishing year.

### 2.3. Selection of Studies

Two authors (J.J.B. and M.H.G.) independently assessed the titles and abstracts of the registers. Relevant studies were chosen, and the full text was searched for further analysis. With the help of a third author, differences in selection were rectified (JCA). The studies were saved in the Endnote 20 software (Clarivate™, Philadelphia, PA, USA).

### 2.4. Outcomes

The primary outcomes were mortality, need for mechanical ventilation, and incidence of pneumothorax. Secondary outcomes were incidence of pneumothorax, APGAR at one and five minutes, umbilical cord pH, and length of hospitalization (days). We used the definitions given by the authors of each RCT.

### 2.5. Data Extraction

Two authors (JJB and MHG) independently extracted data using a pre-developed standardized format. Disagreements were resolved by consensus between the extracting authors, and a third author (JCA) was consulted if necessary. Data extracted per study were author name, year, type of study, country, number of participants, type of intervention, type of control, birth weight, gestational age, APGAR at one and five minutes, umbilical cord pH, length of hospitalization (days), pneumothorax, need for ventilation, and mortality.

### 2.6. Bias Risk Assessment

The following items were assessed for each RCT with the RoB 2.0 Cochrane risk of bias assessment tool [9]: random sequence generation, allocation concealment, blinding of participants and staff, blinding of outcome assessment, blinding and incomplete outcome data, selective reporting, intention to treat, and number of participants excluded from outcome assessment. The reviewers (J.J.B. and J.C.A.) independently assessed the risk of bias by classifying each item separately as low, unclear, or high risk of bias according to the recommendations of Higgins et al. [9].

### 2.7. Statistical Analysis

All meta-analyses used random-effects models and the inverse variance approach. We employed the PM random-effects model because of the expected lack of events. For binary and continuous outcomes, the effect was reported using relative risks (RR) and mean differences (MD) with 95% confidence intervals (95%CI). The I^2^ statistic was used to assess study heterogeneity: 0–30% equaled low heterogeneity, 30–60% equaled moderate heterogeneity, and >60% equaled high heterogeneity. Because of the expected sparsity of events per arm (i.e., a 10% incidence of dichotomous outcomes), we additionally applied sensitivity analyses for primary outcomes using fixed-effect models and the Mantel–Haenzel method. The metabin and metacont functions from the R 3.5.1 statistics package’s meta-library were utilized.

## 3. Results

### 3.1. Selection of RCTs

A total of 446 abstracts were evaluated (Figure 1). A total of 62 duplicate articles were excluded, and of the remaining, 368 were eliminated at the title and abstract stage. Sixteen full-text articles were evaluated, and three RCTs were included in this systematic review [10,11,12] (Figure 1).

### 3.2. Characteristics of Included RCTs

The total number of individuals evaluated was 432 (Table 1). All RCTs were developed in India and evaluated the efficacy of CPAP versus ETI for the resuscitation of neonates with MAS. All studies reported that CPAP and ETI were performed at birth. In all studies, a 48 h follow-up was performed to assess recovery and ensure ventilation in both groups. The mean gestational age in the studies was 38.5 weeks (95%CI: 29.5–47.5). The mean birth weight in the studies was 2863 g (95%CI: 1809.6–3916.4). Only two RCTs [11,12] extracted primary and secondary outcomes. For Chettri [10], only primary outcome information could be extracted. The studies reported that the CPAP intervention started with the CPAP bubble generator and a positive end-expiratory pressure (PEEP) of 5 cm H_2_O.

### 3.3. Risk of Bias in the Included RCTs

Three studies [10,11,12] were scored as having a high risk of bias. These RCTs showed a high risk in the randomization process.

### 3.4. Effect of CPAP on Outcomes

Regarding primary outcomes, no significant differences were found in mortality between both groups (RR = 0.82; 95%CI = 0.54–1.25; I^2^ = 71%; *p* = 0.36; Figure 2a). There was also no significant difference in the need for ventilation between both groups (RR = 0.49; 95%CI = 0.15–1.56; I^2^ = 71%; *p* = 0.57; Figure 2b). There was also no significant difference in the incidence of pneumothorax in patients between both groups (RR = 1.24; 95%CI = 0.30–5.12; I^2^ = 0%; *p* = 0.77; Figure 2c). Regarding secondary outcomes, compared to the ETI group, no significant differences were found in APGAR at one minute (MD = −1.01; 95%CI −2.97 to 0.94; I^2^ = 98%; *p* = 0.31; Figure 3a), APGAR at 5 min (MD = −1.00; 95%CI = −2.96 to 0.95; I^2^= 99%; *p* = 0.32; Figure 3b), days of hospitalization (MD = −0.52; 95%CI = −1.46 to 0.42; I^2^ = 94%; *p* = 0.28; Figure 3c), and cord pH (MD = 0.003; 95%CI = −0.01 to 0.02; I^2^ = 0%; *p* = 0.79; Figure 3d).

## 4. Discussion

We found that CPAP did not reduce mortality, the need for ventilation, or episodes of pneumothorax compared to ETI in neonates with MAS, and there were no significant differences in secondary outcomes. Our study shows that performing CPAP in neonates with MAS does not have a clinically significant benefit compared to performing ETI.

Current recommendations in the management of patients with MAS indicate being the least invasive, with the purpose of ensuring ventilation without damage to the airway anatomy and avoiding consequent collateral effects [13]. There are studies that have compared ventilation therapies with continuous PPV. For example, Bouziri A. et al. [14] developed an RCT with 17 neonates, in which they reported that high-frequency oscillatory ventilation (FiO_2_, MD ± DE = 0.93 ± 0.11) was more effective in reducing invasive ventilation requirements compared to OTI in patients with MAS (FiO_2_, MD ± DE = 0.78 ± 0.25; *p* = 0.031). A systematic review evaluated RCTs comparing ETI and airway suctioning at birth with routine resuscitation, including oropharyngeal suctioning, in vigorous neonates, showing that ETI was not superior (RR = 1.49; 95%CI = 0.86–2.60) [15]. Taking this premise, ETI should not be considered as a principle of airway safety during resuscitation in patients with MAS.

Despite the lack of evidence, the studies addressed considered comparing the efficacy of CPAP versus ETI as ventilation measures in neonates with MAS [16]. The goal of preferring to provide CPAP over ETI was stated to avoid further post-intubation comorbidities. These side effects include the need for mechanical ventilation, pneumothorax, deterioration of neurological function, increased days of hospitalization, among others [17]. Therefore, according to the results, the studies included in this review provide details on the efficacy of CPAP in neonates with MAS. For example, Pandita A. et al. [12] conducted an RCT with 135 participants with a gestational age of 38.2 weeks (SD = 1.3) who had MAS and were not vigorous neonates.

This study shows a high risk of the need for mechanical ventilation in neonates who received CPAP compared to patients who had positive pressure ventilation (OR = 10.67; 95%CI 2.68–71.12; *p* ≤ 0.001). Regarding days of hospitalization, this study showed that patients who received CPAP had a shorter hospitalization time than intubated patients (MD = 4.0; 95%CI = 4.0–6.0 vs. MD = 5.0; 95%CI = 4.0–8.8). However, the RCT by Chettri S. [10] was not able to confirm the efficacy of CPAP because these values were not statistically significant (RR = 1.14; 95%CI = 0.47–1.63; *p* = 0.58). Nevertheless, the authors did not minimize the importance of CPAP and concluded that in the current practice of routine endotracheal suctioning for non-vigorous neonates born with MAS, it should be further evaluated.

Different results were obtained from the RCT by Nangia S. et al. [11], where 175 participants between 37 and 41 weeks of gestational age were evaluated with MAS. The RCT reported a non-significant reduction in the need for ventilation in CPAP patients (OR = 0.85; 95%CI = 0.6–1.4; *p* = 0.67) and no association with the incidence of pneumothorax (OR = 0.90; 95%CI = 0.13–7.1; *p* = 0.45). However, in the relative frequency reported as a function of the need for mechanical ventilation, the proportion of patients with CPAP was lower than that in the ETI group, even though this was not statistically significant (62.5% vs. 19.54%; *p* = 0.67). The length of hospitalization in the group receiving CPAP was not statistically significant compared to the ETI group (MD ± DE = 2.95 ± 0.86 vs. 2.99 ± 1.26, *p* = 0.42). Despite these differences, we decided to perform a meta-analysis because the proportions of adverse outcomes were lower in the group receiving CPAP in all studies.

The American Academy of Pediatrics reported that the presence of meconium fluid might indicate fetal distress and an increased risk of resuscitation [18]. However, there are two specific indications: the first indicates that in vigorous neonates, only the suction bulb should be used to aspirate the oral cavity [19]. The second indicator highlights that in non-vigorous neonates, PPV should be considered, but the American Academy of Pediatrics recommended avoiding ETI a priori to connect to a mechanical ventilator.

Although it is apparently imperative to use CPAP in vigorous and non-vigorous neonates with MAS, the experimental evidence has mainly been from animal models. Consequently, Karlson K. et al. [20] studied positive frequency pressure in an experimental animal model with meconium aspiration, reporting that there was no significant difference between PPV use and mechanical ventilation; therefore, its use should be encouraged when the airway has been secured.

Although the limited evidence on the comparison between CPAP and ETI in neonates with MAS is heterogeneous, the studies share several aspects, such as population, intervention, and control characteristics, and the results are consistent. However, the number of participants varied greatly within each study considered in this review.

The main limitation in this study was the small number of RCTs; statistical heterogeneity was high, and the risk of bias in some domains was uncertain or high. These aspects may alter the interpretation of meta-analyses. In addition, a limitation of the population is that the studies did not report the severity of MAS, which could alter the analysis. However, this is the first meta-analysis to synthesize the evidence comparing both procedures, and, based on the results, we suggest new experimental and more extensive sample size studies.

## 5. Conclusions

In patients with MAS, there is no significant effect of CPAP use compared to ETI on primary and secondary outcomes. Therefore, it is not possible to recommend its general use in clinical practice because the efficacy on primary outcomes, such as mortality, need for ventilation, and reduction in the incidence of pneumothorax, remains uncertain. In addition, more RCTs of better methodological quality and with a larger sample size are needed to learn of effects without a high risk of bias.

## Figures and Tables

**Figure 1 children-09-00589-f001:**
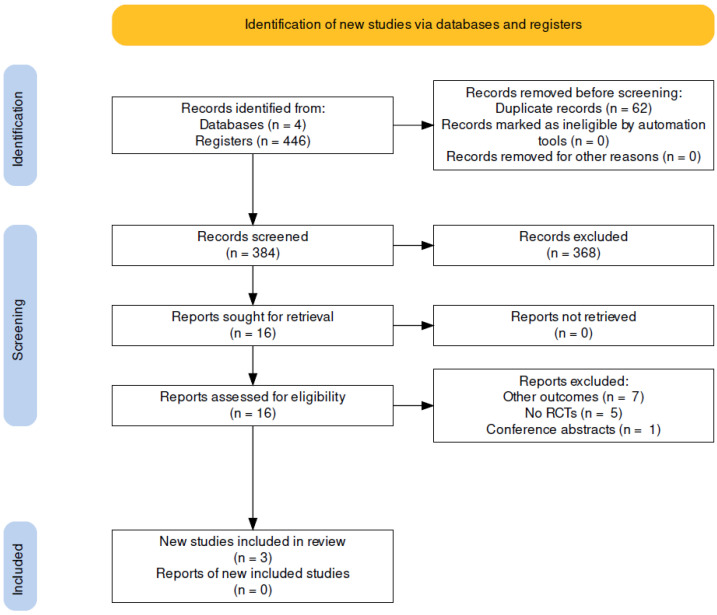
PRISMA flow chart.

**Figure 2 children-09-00589-f002:**
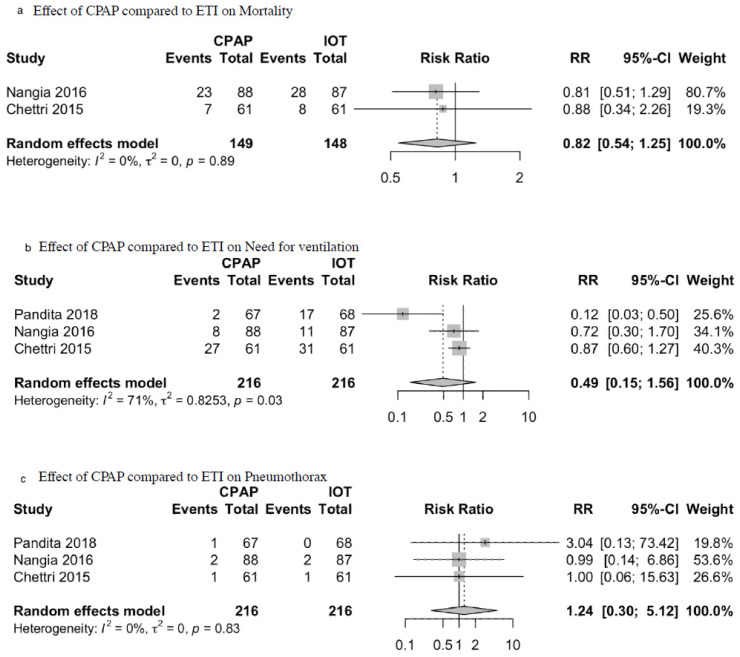
Effect of CPAP compared to ETI on primary outcomes: (**a**) Mortality; (**b**) Need for ventilation; (**c**) Pneumothorax.

**Figure 3 children-09-00589-f003:**
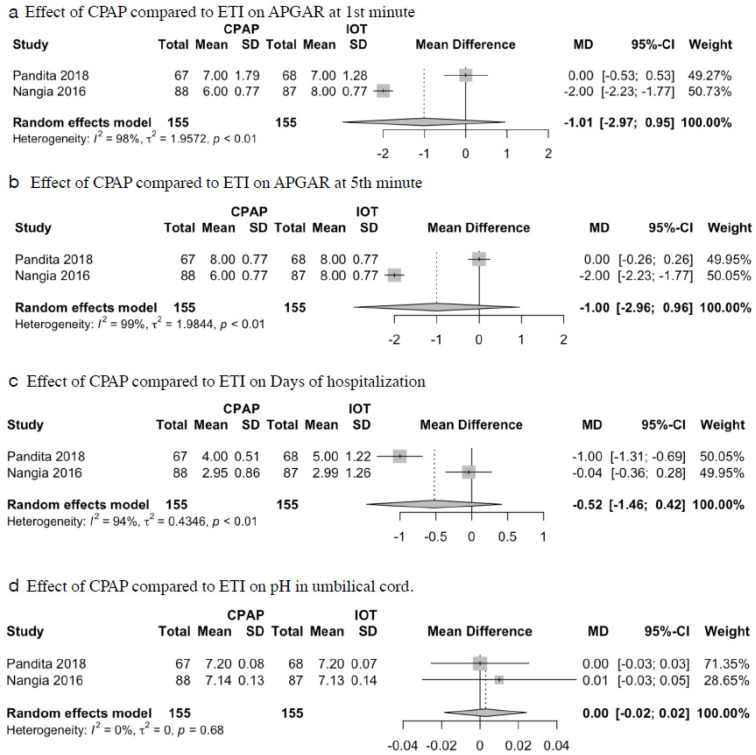
Effect of CPAP compared to ETI on secondary outcomes: (**a**) APGAR at 1st minute; (**b**) APGAR at 5th minute; (**c**) Days of hospitalization; (**d**) pH in umbilical cord.

**Table 1 children-09-00589-t001:** Characteristics of the randomized controlled trials included in the systematic review.

Author	Year	Type of Study	Country	Number of Participants	Interventión	Controls (*n*)	Birth Weight, gm (Mean and sd) (Cpap/Intubatión)	Gestational Age, Weeks (Mean and sd) (Cpap/Intubatión)	Apgar 1 min (Mean and sd) (Cpap/Intubatión)	Apgar 5 min (Mean and sd) (Cpap/Intubatión)	Ph Cord (Mean and sd) (Cpap/Intubatión)	Length of Hospitalization, Days (Mean and sd) (Cpap/Intubatión)	Mortality (ee/te; ec/tc)	Pneumothorax (ee/te; ec/tc)	Ventilation Requeriment (ee/te; ec/tc)
NANGIA [11]	2016	RCT	India	175	No intubation, PPV	Endotracheal intubation	2763 (533)/2649 (437)	39 (0.77)/39 (0.77)	6 (0.77)/8 (0.77)	6 (0.77)/8 (0.77)	7.14 (0.13)/7.13 (0.14)	2.95 (0.86)/2.99 (1.26)	23/88; 28/87	2/88; 2/87	8/88; 11/87
PANDITA [12]	2018	RCT	India	135	CPAP	Endotracheal intubation	2926 (389)/2963 (426)	38.1 (1.3)/38.2 (1.3)	7 (1.79)/7 (1.28)	8 (0.77)/8 (0.77)	7.2 (0.08)/7.2 (0.07)	4 (0.51)/5 (1.22)		1/67; 0/68	2/67; 17/68
CHETTRI [10]	2015	RCT	India	122	No intubation	Endotracheal intubation	2900 (350)/2870 (490)								27/61; 31/61

EE = Events in experiments, TE = Total events, EC = Events in controls, TC = Total controls.

## Data Availability

Not applicable.

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
