# Peer review of "The Efficacy of CPAP in Neonates with Meconium Aspiration Syndrome: A Systematic Review and Meta-Analysis"

_children, 2022, doi:10.3390/children9050589_

Round 1

Reviewer 1 Report

  1. What does SALAM stand for? It needs to be elaborated before using it.
  2. Typo in line 42
  3. Line 44-45-could not find "reducing mortality and bronchopulmonary dysplasia, and without significant increase in severe interventricular hemorrhage" in the reference number 7. Moreover, the mean GA for the RCTs included was 38.5 weeks, in whom we rarely see  bronchopulmonary dysplasia, and severe interventricular hemorrhage.
  4. The primary outcome in the protocol on PROSPERO is different than the actual study
  5. Line 115- What does CPAP8 mean when "positive end-expiratory pressure (PEEP) of 5 cm H 2O"
  6. The RCTs included have significant heterogeneity, > 60%. The high risk of bias in randomization is not clearly explained. 
  7. The outcome of decreased length of hospital stay in CPAP group does not have a significant p value. Moreover, given the high risk of bias in randomization, the outcome of decreased length of hospital stay in CPAP group, could have been due to the fact that infant who were sicker were in the IOT group, ending up with longer length of stay.

Author Response

Our answer in detail: 

1. What does SALAM stand for? It needs to be elaborated before using it.

R: We have updated the term "SALAM" to MAS. 

2. Typo in line 42

R: Thanks, we have corrected.

3. Line 44-45-could not find "reducing mortality and bronchopulmonary dysplasia, and without significant increase in severe interventricular hemorrhage" in the reference number 7. Moreover, the mean GA for the RCTs included was 38.5 weeks, in whom we rarely see  bronchopulmonary dysplasia, and severe interventricular hemorrhage.

R: Thanks for your comments. We have corrected the reference number 7. Regarding the second point, it should be noted that BPD and IVH are complications reported in patients with SAM, which have been reported by several studies. We are describing what other studies have mentioned. 

4. The primary outcome in the protocol on PROSPERO is different than the actual study

R: Definitely. The change in the concept of the primary outcome is due to the finding of the primary outcomes found in the RCTs. Thanks for the observation. Considered that the outcomes predicted in the manuscript do not differ from those proposed in the protocol.

5. Line 115- What does CPAP8 mean when "positive end-expiratory pressure (PEEP) of 5 cm H 2O"

R: Thanks. We have corrected this sentence.

6. The RCTs included have significant heterogeneity, > 60%. The high risk of bias in randomization is not clearly explained. 

R: Thanks for your comments. Heterogeneity can be due to many factors. Among them, the number of studies, the difference between the sample size per arm within each study, and the differences between the effects of the outcomes. In our risk of bias analysis, we mention only that there is a high risk of bias due to randomization, but that does not explain the heterogeneity.

7. The outcome of decreased length of hospital stay in CPAP group does not have a significant p value. Moreover, given the high risk of bias in randomization, the outcome of decreased length of hospital stay in CPAP group, could have been due to the fact that infant who were sicker were in the IOT group, ending up with longer length of stay.

R: Thanks a lot. We have corrected all analysis for secondary outcomes, and changed the effect measures and 95%CIs. 

Reviewer 2 Report

Overall this systematic review addresses an important clinical question within the delivery room management of neonates born through meconium stained amniotic fluid: what is the role of CPAP.  It has always been concerning that CPAP, in these cases, may place infants at risk of pneumothorax when compared to mother modes of respiratory support or mechanical ventilation.  It was very interesting to read that CPAP had no  significant effect on the primary outcomes chosen, which I think is very valuable information. However, the significant decrease in the 1 minute APGAR score found in secondary analysis is somewhat concerning, and while statistically significant, I am not sure that the decrease in length of stay by 0.53 days is of clinical significance.  This topic was ripe for a systematic review, and I agree with your conclusion that this topic could use more clinical trial investigation.  I do believe that this review speaks to the notion that it could, in fact, be ethical to perform such trials, given that there was no detrimental effect seen with CPAP in any of the primary outcomes of death, need for mechanical ventilation, or pneumothorax.

The biggest limitations I see with this paper itself are in some of the definitions of abbreviations used.  "SALAM" and "IOT" are used throughout the manuscript and are never defined.  I was able to deduce the meaning of SALAM (which is also described in other places as "MAS" which is more common in the American literature, however IOT remained illusive to me, especially since "orotracheal intubation" had another abbreviation that was, in fact, defined.  I am interpreting IOT to mean the same as orotracheal intubation, for the purpose of the review, but if it is not, I am afraid the premise of the comparison group may have been lost on me.

NRP guidelines have changed with regard to the delivery room management of infants born through meconium stained amniotic fluid during the period which the included articles were written.  It would be helpful to note this in the manuscript, as it does certainly affect the methods and outcomes of those individual studies.  

Finally, it appears to me that Table 1 could used some formatting edits, and it would be easier to interpret if figures 2 and 3 had each plot labeled individually rather than in the caption for the whole figure.

Author Response

Dear Reviewer.

Thanks for your comments. We have corrected and changed based on your observations. 
